# Neural Symbolic Reader: Scalable Integration of Distributed and Symbolic Representations for Reading Comprehension

**Xinyun Chen** [*]
UC Berkeley
xinyun.chen@berkeley.edu

**Chen Liang, Adams Wei Yu, Denny Zhou**
Google Brain
{crazydonkey,adamsyuwei,dennyzhou}@google.com

**Dawn Song**
UC Berkeley
dawnsong@cs.berkeley.edu

**Quoc V. Le**
Google Brain
qvl@google.com

## Abstract

Integrating distributed representations with symbolic operations is essential for reading comprehension requiring complex reasoning, such as counting, sorting and arithmetics, but most existing approaches rely on specialized neural modules and are hard to adapt to multiple domains or multi-step reasoning. In this work, we propose the **Ne**ural Symbolic **Rea**der (**NeRd**), which includes a *reader*, e.g., BERT, to encode the passage and question, and a *programmer*, e.g., LSTM, to generate a program for multi-step reasoning. By using operators like span selection, the program can be executed over text to generate the answer. Compared to previous works, NeRd is more *scalable* in two aspects: (1) *domain-agnostic*, i.e., the same neural architecture works for different domains; (2) *compositional*, i.e., complex programs can be generated by compositionally applying the symbolic operators. Furthermore, to overcome the challenge of training NeRd with weak supervision, we apply data augmentation techniques and hard Expectation-Maximization (EM) with thresholding. On DROP, a challenging reading comprehension dataset requiring discrete reasoning, NeRd achieves 1.37%/1.18% absolute gain over the state-of-the-art on Exact-Match/F1 metrics. With the same architecture, NeRd significantly outperforms the baselines on MathQA, a math problem benchmark that requires multiple steps of reasoning, by 25.5% absolute gain on accuracy when trained on all the annotated programs, and more importantly, still beats the baselines even with only 20% of the program annotations.

## 1 Introduction

Deep neural networks have achieved remarkable successes in natural language processing recently. In particular, pretrained language models, e.g., BERT (Devlin et al., 2019), have significantly advanced the state-of-the-art in reading comprehension. While neural models have demonstrated performance superior to humans on some benchmarks, e.g., SQuAD (Rajpurkar et al., 2016), so far such progress is mostly limited to extractive question answering, in which the answer is a single span from the text. In other words, this type of benchmarks usually test the capability of text pattern matching, but not of reasoning. Some recent datasets, e.g., DROP (Dua et al., 2019) and MathQA (Amini et al., 2019), are collected to examine the capability of both language understanding and discrete reasoning, where the direct application of the state-of-the-art pre-trained language models, such as BERT or QANet (Yu et al., 2018), achieves very low accuracy. This is especially challenging for pure neural network approaches, because discrete operators learned by neural networks, such as addition and sorting, can hardly generalize to inputs of arbitrary size without specialized design (Reed & de Freitas, 2016; Cai et al., 2017; Kaiser & Sutskever, 2015). Therefore, integrating neural networks with symbolic reasoning is crucial for solving those new tasks.

The recent progress on neural semantic parsing (Jia & Liang, 2016; Liang et al., 2017) is sparked to address this problem. However, such success is mainly restricted to question answering with structured data sources, e.g., knowledge graphs (Berant et al., 2013) or tabular databases (Pasupat

---

[*]Work was done while interning at Google Brain.

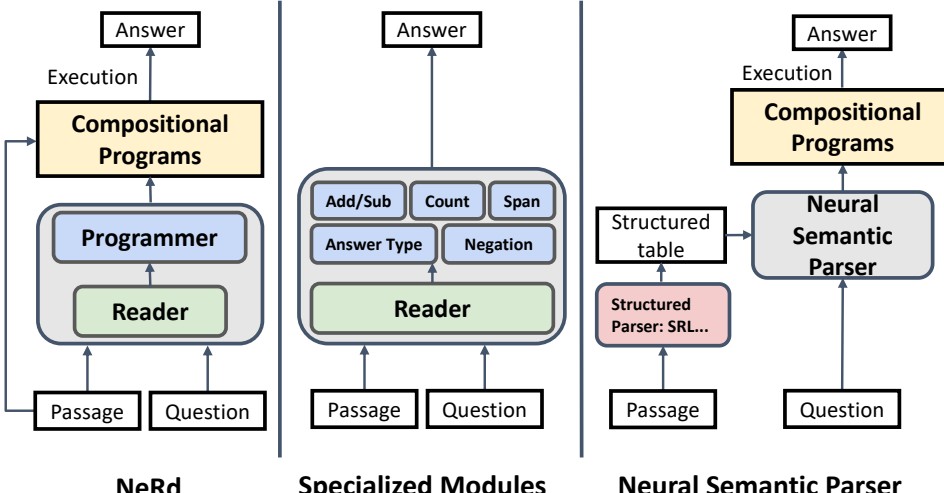

Figure 1: Comparison of NeRd with previous approaches for reading comprehension requiring complex reasoning. The components in grey boxes are the neural architectures. Previous works mainly take two approaches: (1) augmenting pre-trained language model such as BERT with specialized modules for each type of questions, which is hard to scale to multiple domains or multi-step complex reasoning; (2) applying neural semantic parser to the structured parses of the passage, which suffers severely from the cascade error. In contrast, the neural architecture of NeRd is domain-agnostic, which includes a *reader*, e.g., BERT, and a *programmer*, e.g., LSTM, to generate compositional programs that are directly executed over the passages.

& Liang, 2015). Extending it to reading comprehension by parsing the text into structured representations suffers severely from the cascade errors, i.e., the issues of the structured parsing for data preprocessing account for the poor performance of the learned neural model (Dua et al., 2019).

A recent line of work (Dua et al., 2019; Hu et al., 2019; Andor et al., 2019) extends BERT/QANet to perform reasoning on the DROP dataset. However, they cannot easily scale to multiple domains or multi-step complex reasoning because: (1) they usually rely on handcrafted and specialized modules for each type of questions; (2) they don't support compositional applications of the operators, so it is hard to perform reasoning of more than one step.

In this work, we propose the **Ne**ural Symbolic **R**ea**d**er (NeRd) for reading comprehension, which consists of (1) a *reader* that encodes passages and questions into vector representations; and (2) a *programmer* that generates programs, which are executed to produce answers. The key insights behind NeRd are as follows: (1) by introducing a set of span selection operators, the compositional programs, usually executed against structured data such as databases in semantic parsing, can now be executed over text; (2) the same architecture can be applied to different domains by simply extending the set of symbolic operators.

A main challenge of training NeRd is that it is often expensive to collect program annotations, so the model needs to learn from weak supervision, i.e., with access only to the final answers. This raises two problems for learning: (1) cold start problem. There are no programs available at the beginning of training, so the training cannot proceed. We address this problem through data augmentation that generates noisy training data to bootstrap the training; (2) spurious program problem, where some programs produce the right answer for wrong rationales. We propose an iterative process using hard EM with thresholding, which filters out the spurious programs during training.

In our evaluation, NeRd demonstrates three major advantages over previous methods: (1) better accuracy. It outperforms the previous state-of-the-art on DROP by 1.37%/1.18% on EM/F1, and the baselines on MathQA by a large margin of 25.5% on accuracy if trained with all annotated programs. Notably, it still outperforms the MathQA baselines using only 20% of the program annotations; (2) more scalable (domain-agnostic and compositional). Unlike previous approaches, which rely on specialized modules that do not support compositional application of the operators, NeRd can be applied to tasks of different domains, e.g., DROP and MathQA, without changing the architecture, and more complex programs can be simply generated by extending the set of operators and compo-

sitionally applying them; (3) better interpretability. It is easier to interpret and verify an answer by inspecting the program that produces it, especially for the questions involving complex reasoning such as counting and sorting.

## 2 NEURAL SYMBOLIC READER

In this section, we present the design of NeRd. It consists of a *reader* that encodes the passages and questions into vector representations, and a *programmer* that generates programs in a domain specific language. The overall comparison between NeRd and previous works is visualized in Figure 1.

### 2.1 NEURAL ARCHITECTURE

We provide an overview of the two components in NeRd, and defer more details to Appendix C.

**Reader.**    Given the natural language text including a question and a passage, the reader component encodes each token $t_i$ in the text into an embedding $e_i$. Note that our framework is agnostic to the architecture choice of the encoder, so any neural module that turns words into vectors is applicable, e.g., BERT (Devlin et al., 2019).

**Programmer.**    The programmer takes the output of the reader as input, and then decodes a program as a sequence of tokens. Again, our model is agnostic to the design of decoder. For simplicity, we use an LSTM (Hochreiter & Schmidhuber, 1997) decoder with attention (Bahdanau et al., 2014) over the encoded text, and self-attention (Vaswani et al., 2017) over the previously generated tokens.

A major advantage of our architecture is that it is *domain-agnostic*, i.e., the same architecture can be used for different domains. Compared to previous approaches that craft separate specialized modules for each answer type, we use a unified programmer component to generate programs for multi-step reasoning, and we can simply extend the operator set in the domain specific language (see the next section) to adapt to a different domain. See Section 4.3 for a more detailed discussion.

### 2.2 DOMAIN SPECIFIC LANGUAGE

In this section, we introduce our domain specific language (DSL), which is used to interpret the tokens generated by the programmer component as an executable program.

We list the operators in our DSL in Table 1. To handle discrete reasoning, the DSL includes operators that perform arithmetics (`DIFF`, `SUM`), counting (`COUNT`) and sorting (`ARGMAX`, `ARGMIN`, `MAX`, `MIN`). These operators have been used in previous work in semantic parsing over structured data sources such as a knowledge graph or a tabular database.

However, the main challenge of applying such operations for reading comprehension is that the model needs to manipulate unstructured data, i.e., natural language text, and parsing the text into structured representations may introduce a lot of cascade errors. For example, Dua et al. (2019) found that their best performing semantic parsing pipeline using SRL (Carreras & Màrquez, 2004) can only find the logical forms for 35% of the questions, resulting in poor performance.

To address this issue, a key insight in our DSL design is to introduce the span selection operators, so that all the arithmetics, counting and sorting operators can be applied to text. Specifically, we introduce `PASSAGE_SPAN`, `QUESTION_SPAN`, `VALUE`, `KEY-VALUE` for selecting spans or numbers from the passage and question. For example, `COUNT` can use `PASSAGE_SPAN` to pick out the spans that mention the relevant entities or events, e.g., touchdowns made by a certain person, and then returns the total number; `ARGMAX` relies on applying `KEY-VALUE` to pick out the spans (keys) for relevant mentions and their associated numbers (values), e.g., touchdowns and their lengths, and then returns the key with the highest value, e.g., the player kicking the longest touchdown. More examples can be found in Table 2. In summary, the introduction of span selection operators in the DSL enables the application of the discrete reasoning operators to text, and the resulting programs act as executable and interpretable representations of the reasoning process.

As mentioned above, our architecture is domain-agnostic and the only change needed, to apply to a different domain, is to extend the DSL with new operators. For example, MathQA benchmark requires adding more advanced mathematical operations beyond addition and subtraction, which are defined in Amini et al. (2019). We defer the details to Section 4.1.

A major advantage of our DSL is its *compositionality*, i.e., complex programs can be generated by compositionally applying the operators. Previous works (Andor et al., 2019) only allow applying the operators for one step, which requires them to introduce operators to mimic two-step compositions,

| Operator | Arguments | Outputs | Description |
|---|---|---|---|
| PASSAGE_SPAN QUESTION_SPAN | **v0**: the start index. **v1**: the end index. | a span. | Select a span from the passage or question. |
| VALUE | **v0**: an index. | a number. | Select a number from the passage. |
| KEY-VALUE (KV) | **v0**: a span. **v1**: a number. | a key-value pair. | Select a key (span) value (number) pair from the passage. |
| DIFF SUM | **v0**: a number or index. **v1**: a number or index. | a number. | Compute the difference or sum of two numbers. |
| COUNT | **v**: a set of spans. | a number. | Count the number of given spans. |
| MAX MIN | **v**: a set of numbers. | a number. | Select the maximum / minimum among the given numbers. |
| ARGMAX ARGMIN | **v**: a set of key-value pairs. | a span. | Select the key (span) with the highest / lowest value. |

Table 1: Overview of our domain-specific language. See Table 2 for the sample usage.

e.g., `Merge` (selecting two spans) and `Sum3` (summing up three numbers). However, this would not scale to more steps of reasoning, as the number of required operators will grow exponentially w.r.t the number of steps. In contrast, NeRd can compose different operators to synthesize complex programs for multi-step reasoning. For example, on MathQA, the average number of operations per question is 5, and some programs apply more than 30 operations to compute the final answer.

# 3 TRAINING WITH WEAK SUPERVISION

Although it is relatively easy to collect question-answer pairs, it is often hard and expensive to obtain program annotations that represent the reasoning behind the answers. Thus, how to train NeRd with only weak supervision becomes a main challenge. In this section, we revisit the cold start and spurious program problems described in Section 1, and present our solutions.

## 3.1 DATA AUGMENTATION FOR COLD START

The cold start problem means that the training cannot get started when there isn't any program available. For example, a question "How many touchdowns did Brady throw" annotated with only an answer "3" cannot be directly used to train our model due to the lack of the target program to optimize on. To obtain program annotations from question-answer pairs, we first follow previous work to find programs for questions answerable by span selection or arithmetic operations via an exhaustive search, and we defer the details to Section 4.2. However, for questions involving counting or sorting operations, the space becomes too large for an exhaustive search, since these operations rely on the span selection as their sub-routines. For example, the number of possible spans in a text with 200 words is in the order of $10^4$, and what's more, counting and sorting operators usually include more than one span as their arguments.

We apply data augmentation to address the search space explosion problem for counting and sorting operations. For counting, we augment the span selection questions by replacing the interrogatives, e.g., "what" and "who", with "how many" when applicable, and adding a call to COUNT over the selected spans in the answer. For example, a question "What areas have a Muslim population of more than 50000 people?" is changed into "How many areas...". For sorting, we extract the key-value pairs by first applying CoreNLP (Manning et al., 2014) for entity recognition, and then heuristically find an associated number for each entity. If including them as the arguments of any sorting operator yields the correct answer, then such programs are added to the training set. More details can be found in Appendix D.1. Although the programs found for counting and sorting through this data augmentation process is noisy, they help bootstrap the training. Throughout the training, we also use the model to decode programs, and add those leading to correct answers into our training set.

## 3.2 HARD EM WITH THRESHOLDING AGAINST SPURIOUS PROGRAMS

After collecting a set of programs for each question-answer pair, another obstacle is the spurious program problem, the phenomenon that a wrong program accidentally predicts a right answer. For example, per arithmetic question in DROP, there are on average $9.8$ programs that return correct answers, but usually only one of them is semantically correct.

---

**Algorithm 1** Hard EM with Thresholding

---

**Input:** question-answer pairs $\{(x_i, y_i)\}_{i=1}^N$,
a model $p_\theta$, initial threshold $\alpha_0$, decay factor $\gamma$
**for** each $(x_i, y_i)$ **do**
    $Z_i \leftarrow \text{DataAugmentation}(x_i, y_i)$
$T \leftarrow 0$
**repeat**
    $\alpha \leftarrow \alpha_0 * \gamma^T$
    $\mathcal{D} \leftarrow \emptyset$
    **for** each $(x_i, y_i)$ **do**
        $z_i^* = \arg\max_k p_\theta(z_i^k | x_i), z_i^k \in Z_i$
        **if** $p_\theta(z_i^*) > \alpha$ or $T = 0$ and $|Z_i| = 1$ **then**
            $\mathcal{D} \leftarrow \mathcal{D} \cup (x_i, z_i^*)$
    Update $\theta$ by maximizing $\sum_\mathcal{D} \log p_\theta(z^* | x)$
    $T \leftarrow T + 1$
**until** converge or early stop

---

To filter out spurious programs, we adopt hard EM (Liang et al., 2018; Min et al., 2019) due to its simplicity and efficiency. Specifically, this approach uses the current model to select the program with the highest model probability among the ones that return the correct answer, and then maximizes the likelihood of the selected program. In other words, it relies on the neural model itself to filter out spurious programs. This algorithm is usually faster than the marginalized approach (Berant et al., 2013) because at most one program per question-answer pair is used to compute the gradient, and the selection process is fast since it only has a forward pass.

Hard EM assumes that for any question-answer pair, at least one of the generated programs is correct. However, there exist questions without any semantically correct program found, e.g., when the annotated answer itself is wrong. In this case, when directly applying the hard EM algorithm, even if the model probabilities for all the programs are very small, it will still select a program for training. RL-based approaches such as MAPO (Liang et al., 2018) avoid this issue by optimizing the expected return, which weighs the gradient by the model probability. Thus, when all the programs of a question-answer pair have very small probabilities, they will be largely ignored during training. We incorporate this intuition into hard EM by introducing a decaying threshold $\alpha$, so that a program's probability has to be at least $\alpha$ in order to be included for training. Our experiments show that both hard EM and thresholding are crucial for successful training. The pseudo-code of our training procedure is presented in Algorithm 1, and we defer more details to Appendix D.2.

## 4 EVALUATION

In this section, we demonstrate the effectiveness of our approach on DROP (Dua et al., 2019) and MathQA (Amini et al., 2019), two recent benchmarks that require discrete reasoning over passages.

### 4.1 DATASETS

**DROP.** DROP (Discrete Reasoning Over Paragraphs) (Dua et al., 2019) is designed to combine the challenges from both reading comprehension and semantic parsing communities. Specifically, the passages are collected from Wikipedia, each having at least twenty numbers. The question-answer pairs are crowdsourced in an adversarial way that they are accepted only when the questions cannot be correctly answered by the BiDAF model (Seo et al., 2017). The dataset has 96.6K question-answer pairs from 6.7K passages. Unlike most existing datasets that are solely based on the single span selection, the questions in DROP require complex reasoning, such as selecting multiple spans, arithmetic operations over numbers in the passage, counting and sorting, etc., which poses extra challenge for existing models. For example, vanilla BERT only gets around 30% F1 score. Table 2 provides some sample questions in DROP, and their corresponding programs in our DSL (Table 1).

For evaluation, we use the same metrics in Dua et al. (2019): (1) Exact Match (EM), where the score is 1 if the prediction exactly matches the ground truth, and 0 otherwise; (2) F1 score, which gives partial credits to a prediction that is not exactly the same as the ground truth, but overlaps with it.

**MathQA.** MathQA (Amini et al., 2019) is a dataset with 37K question-answer pairs selected from AQuA (Ling et al., 2017), but it is further annotated with gold programs in their domain-specific language. The passage length in MathQA is 38 on average, much shorter than DROP with 224. However, the questions in MathQA require more complex and advanced mathematical reasoning than DROP. To this aim, they design 58 math operations, which cover various advanced math topics including geometry, physics, probability, etc. Accordingly, we augment our DSL with those operators to support more advanced numerical reasoning. In these annotated programs, the average number of operations per question is 5, and some programs involve more than 30 steps of computation. Table 3 shows an example from MathQA.

| Passage | Question & Answer |
|---------|-------------------|
| Multiple spans | |
| ...the population was spread out with **26.20% under the age of 18**, 9.30% from 18 to 24, **26.50% from 25 to 44**, **23.50% from 45 to 64**, and 14.60% who were 65 years of age or older... | **Question:** Which groups in percent are larger than 16%? **Program:** PASSAGE_SPAN(26,30), PASSAGE_SPAN(46,48), PASSAGE_SPAN(55,57) **Result:** 'under the age of 18', '25 to 44', '45 to 64' |
| Date | |
| When major general Nathanael Greene took command in the south, Marion and lieutenant colonel Henry Lee were ordered in January **1781**... On **August 31**, Marion rescued a small American force trapped by 500 British soldiers... | **Question:** When did Marion rescue the American force? **Program:** PASSAGE_SPAN(71,71), PASSAGE_SPAN(72,72), PASSAGE_SPAN(32,32) **Result:** 'August', '31', '1781' |
| Numerical operations | |
| ...Lassen county had a population of **34,895**. The racial makeup of Lassen county was **25,532 (73.2%) white** (U.S. census), **2,834 (8.1%) African American** (U.S. census)... | **Question:** How many people were not either solely white or solely African American? **Program:** DIFF(9,SUM(10,12)) **Result:** 34895 - (25532 + 2834) = 6529 |
| Counting | |
| ...the Bolshevik party came to power in November 1917 through the **simultaneous election in the soviets** and an **organized uprising supported by military mutiny**... | **Question:** How many factors were involved in bringing the Bolsheviks to power? **Program:** COUNT( PASSAGE_SPAN(62, 66), PASSAGE_SPAN(69, 74)) **Result:** COUNT( 'simultaneous election in the soviets', 'organized uprising supported by military mutiny') = 2 |
| Sorting | |
| ...Jaguars kicker **Josh Scobee** managed to get a 48-yard field goal...with kicker Nate Kaeding getting a 23-yard field goal... | **Question:** Who kicked the longest field goal? **Program:** ARGMAX( KV(PASSAGE_SPAN(50,53),VALUE(9)), KV(PASSAGE_SPAN(92,94),VALUE(11))) **Result:** ARGMAX( KV('Josh Scobee', 48), KV('Nate Kaeding', 23)) = 'Josh Scobee' |
| ...Leftwich flipped a **1-yard touchdown pass** to Wrighster...Leftwich threw a 16- yard touchdown pass to Williams for a 38-0 lead... | **Question:** How many yards was the shortest touchdown pass? **Program:** MIN(VALUE(17), VALUE(19)) **Result:** MIN(1, 16) = 1 |

Table 2: Examples of correct predictions on DROP development set.

| Question | Answer |
|----------|--------|
| Someone on a skateboard is traveling 8 miles per hour. How many feet does she travel in 5 seconds? (1 mile = 5280 feet) | **Program:** multiply(5,divide(multiply(8,5280),const_3600)) **Result:** 5 * ((8 * 5280) / 3600) = 58.67 ft |

Table 3: An example in MathQA dataset.

Note that each question in MathQA is accompanied with 4 options, where 1 of them is the correct answer. However, since we do not have the full knowledge of the operation semantics, we choose a conservative metric to evaluate the accuracy: a predicted program is considered to be correct only if it is exactly the same as the annotated program. Thus, this metric is an under-estimation of the accuracy based on the execution results. Despite that we use a much stricter measurement in our evaluation, we show that NeRd still outperforms the baselines by a large margin.

## 4.2 IMPLEMENTATION DETAILS

**DROP.** Similar to previous work (Dua et al., 2019), for span prediction, we perform an exhaustive search to find all mentions of the ground truth spans in the passage, then include all of them as candidate programs. For numerical questions, we perform another exhaustive search over all expressions applying addition and subtraction over up to 3 numbers. In this way, we are able to find at least one

program for over $95\%$ of the training samples with a number as the answer. Our data augmentation approach for counting and sorting questions can be seen in Section 3.1.

**MathQA.** Besides the setting where all the ground truth programs are provided during training, we also evaluate the weak supervision setting on MathQA. Due to the lack of program executor, we are unable to perform the search similar to what we have done on DROP. To enable the first training iteration of the model, we assume that we have access to the ground truth programs for a small fraction of training samples at the beginning, and only know the final answer for the rest of training samples. In the first training iteration, the model only trains on the samples annotated with programs. In each of the following iterations, we first run a beam search with a beam size 64 to generate programs for each training sample that has not been annotated in previous iterations, and add the generated program only if it is exactly the same as the ground truth annotation.

For a fair comparison, our reader uses the same pre-trained model as (Hu et al., 2019; Andor et al., 2019), i.e., BERT$_{\text{LARGE}}$. For both benchmarks, we perform greedy decoding during the evaluation.

### 4.3 BASELINES

**DROP.** We evaluate NeRd against three types of baselines: (1) previous models on DROP; (2) NeRd with and without counting and sorting operations; (3) NeRd with different training algorithms, and we discuss the details below.

*Previous approaches.* We compare with NAQANet (Dua et al., 2019), NABERT (Hu et al., 2019), MTMSN (Hu et al., 2019), and BERT-Calc (Andor et al., 2019). We have discussed the key differences between NeRd and BERT-Calc, the baseline with the best performance, in Section 2.2. On the other hand, NAQANet, NABERT, MTMSN share the same overall framework, where they augment an existing model to include individual modules for span selection, numerical expression generation, counting, negation, etc. While NAQANet is based on QANet, other baselines as well as NeRd are based on BERT. Note that the span selection modules themselves are not able to handle questions that return multiple spans as the answer, which causes the exact match accuracy to be zero on multiple-span selection questions for both NAQANet and NABERT. To tackle this issue, MTMSN adapts the non-maximum suppression algorithm (Rosenfeld & Thurston, 1971) to select multiple spans from the candidates with the top prediction probabilities.

*Operator variants of NeRd.* To show that NeRd learns to apply counting and sorting operations appropriately, we also evaluate the following two variants: (1) *NeRd without counting*: we remove the `COUNT` operation in Table 1, and introduce 10 operations `COUNT_0`, `COUNT_1`, ..., `COUNT_9`, where the execution engine returns the number $x$ for operation `COUNT_X`. This counting process is the same as (Andor et al., 2019). (2) *NeRd without sorting*: we remove `ARGMAX`, `ARGMIN`, `MAX` and `MIN` operations, so that the model needs to use span selection operations for sorting questions.

*Training variants of NeRd.* To show the effectiveness of our training algorithm, we compare with the following baselines: (1) *Hard EM* described in Section 3.2; and (2) *Maximum Likelihood*, which maximizes the likelihood of each program that returns the correct answer for a training sample.

**MathQA.** We compare with Seq2prog and Seq2prog+cat models in Amini et al. (2019), which are LSTM-based encoder-decoder architectures implemented in OpenNMT (Klein et al., 2018). In particular, Seq2prog+cat extracts the category label of each question, then trains separate LSTMs to handle different categories, which improves the accuracy by $2.3\%$.

### 4.4 RESULTS

**DROP.** Table 4 summarizes our main evaluation results on DROP dataset, with 9.5K samples in the development set and 9.6K hidden samples in the test set. Note that NABERT$_{\text{LARGE}}$ was not evaluated on the test set (Hu et al., 2019). Specifically, we train 10 NeRd models with the best configuration from different random initialization, present the mean and standard error of the results on the development set, and submit a single model to obtain the result on the hidden test set. We can observe that on test set, NeRd outperforms previous models by $1.37\%$ on exact match, and $1.18\%$ on F1 score. Notice that in (Andor et al., 2019), they train their BERT-Calc model on CoQA (Reddy et al., 2019) in addition to DROP, and they also evaluate an ensemble with 6 models, resulting in the exact match of $78.14$, and F1 score of $81.78$ on test set. However, we can see that without additional training data and ensembling, NeRd still beats their single model, and the performance is on par with their ensemble model.

| | Overall Dev | | Overall Test | | Number (62%) | | Span (32%) | | Spans (4.4%) | | Date (1.6%) | |
|---|---|---|---|---|---|---|---|---|---|---|---|---|
| | EM | F1 | EM | F1 | EM | F1 | EM | F1 | EM | F1 | EM | F1 |
| NAQANet | 46.75 | 50.39 | 44.24 | 47.77 | 44.9 | 45.0 | 58.2 | 64.8 | 0.0 | 27.3 | 32.0 | 39.6 |
| NABERT$_{\text{LARGE}}$ | 64.61 | 67.35 | – | – | 63.8 | 64.0 | 75.9 | 80.6 | 0.0 | 22.7 | 55.7 | 60.8 |
| MTMSN$_{\text{LARGE}}$ | 76.68 | 80.54 | 75.85 | 79.85 | 80.9 | 81.1 | 77.5 | 82.8 | 25.1 | 62.8 | 55.7 | 69.0 |
| BERT-Calc | 78.09 | 81.65 | 76.96 | 80.53 | 82.0 | 82.1 | 78.8 | 83.4 | 5.1 | 45.0 | 58.1 | 61.8 |
| NeRd | **78.55** ± 0.27 | **81.85** ± 0.20 | 78.33 | 81.71 | **82.4** ± 0.3 | **82.6** ± 0.2 | 76.2 ± 0.4 | 81.8 ± 0.2 | **51.3** ± 0.8 | **77.6** ± 1.2 | **58.3** ± 1.8 | 67.2 ± 1.7 |

Table 4: Results on DROP dataset. On the development set, we present the mean and standard error of 10 NeRd models, and the test result of a single model. For all models, the performance breakdown of different question types is on the development set. Note that the training data of BERT-Calc model (Andor et al., 2019) for test set evaluation is augmented with CoQA (Reddy et al., 2019).

| | with Count Op | w/o Count op |
|---|---|---|
| EM | **73.1** | 71.2 |
| F1 | **73.1** | 71.2 |

(a)

| | with Sort Ops | w/o Sort Ops |
|---|---|---|
| EM | **83.9** | 82.1 |
| F1 | **86.8** | 85.5 |

(b)

Table 5: Results of counting and sorting questions on DROP development set, where we compare variants of NeRd with and without the corresponding operations. **(a)**: counting; **(b)**: sorting. For each setting, we present the best results on development set.

To understand the strengths of NeRd, we first show examples of correct predictions in Table 2. We can observe that NeRd is able to compose multiple operations so as to obtain the correct answer, which helps boost the performance. In particular, for questions that require the selection of multiple spans, the exact match accuracy of NeRd is more than double of the best previous approach that specially designed for multi-span prediction, and the F1 score also improves around 15%. Meanwhile, NeRd is able to generate more complicated arithmetic expressions than Andor et al. (2019), thanks to the compositionality of our approach.

We further present our ablation studies of counting and sorting operations in Tables 5 and 6. Specifically, we evaluate on two subsets of DROP development set that include counting and sorting questions only, using the variants of NeRd with and without the corresponding operations. We can observe that adding these advanced operations can not only boost the performance, but also enable the model to provide the rationale behind its predictions. For counting problems, NeRd is able to select the spans related to the question. For sorting problems, NeRd first associates the entities with their corresponding values to compose the key-value pairs, then picks the most relevant ones for prediction. None of the previous models is able to demonstrate such reasoning processes, which suggests better interpretability of NeRd.

Finally, we present the results of different training algorithms in Table 7. First, we observe that by filtering spurious programs, the hard EM significantly boosts the performance of the maximum likelihood training for 10%, which may be due to the fact that the exhaustive search finds plenty of spurious programs that yield the correct answer. Adding the threshold for program selection provides further improvement of about 7%, indicating that our training algorithm can better handle the issue of spurious programs and be more tolerant to the noise of answer annotations. In Appendix E, we show some examples discarded by NeRd using the threshold, which mostly have the wrong answer annotations, e.g., incorrect numerical operations or missing part of the information in the question.

**MathQA.** We present the results on MathQA test set with around 3K samples in Table 8. NeRd dramatically boosts the accuracy of the baselines by 25.5%. In addition, we also evaluate a variant of NeRd with the same model architecture, but the BERT encoder is not pre-trained and is randomly initialized. We observe that this variant still yields a performance gain of 17.4%. Note that NeRd is measured by the program accuracy, which is a much stricter criterion and thus is an underestimation of the execution accuracy computed in (Amini et al., 2019). Moreover, even with only 20% training data labeled with ground truth programs, NeRd still outperforms the baseline.

## 5 RELATED WORK

Reading comprehension and question answering have recently attracted a lot of attention from the NLP community. A plethora of datasets have been available to evaluate different capabilities of

| Passage | Question & Prediction |
|---|---|
| ...with field goals of **38** and **36 yards** by kicker Dan Carpenter ... followed by a **43-yard** field goal by Carpenter ... **52-yard** field goal ... | **Question:** How many total field goals were kicked in the game? |
| | **Predicted Program:**
COUNT(
 PASSAGE_SPAN(75,75), PASSAGE_SPAN(77,78),
 PASSAGE_SPAN(133,135), PASSAGE_SPAN(315,317))
**Result:** COUNT( '38','36 yards', '43-yard','52-yard') = 4 |
| | **Predicted Program (-counting):** COUNT5 **Result:** 5 |
| ... with the five most common surgeries being breast augmentation, **liposuction**, breast reduction, **eyelid surgery** and **abdominoplasty** ... | **Question:** How many of the five most common procedures are not done on the breasts? |
| | **Predicted Program:**
COUNT(
 PASSAGE_SPAN(132,135), PASSAGE_SPAN(140,142), PASSAGE_SPAN(144,149))
**Result:** COUNT( 'liposuction', 'eyelid surgery', 'abdominoplasty') = 3 |
| | **Predicted Program (-counting):** COUNT4 **Result:** 4 |

(a)

| Passage | Question & Prediction |
|---|---|
| ...In the third quarter, Arizona's deficit continued to climb as Cassel completed a 76-yard touchdown pass to wide receiver Randy Moss ... quarterback **Matt Leinart** completed a 78-yard touchdown pass to wide receiver Larry Fitzgerald ... | **Question:** Who threw the longest touchdown pass? |
| | **Predicted Program:**
ARGMAX(
 KV(PASSAGE_SPAN(205,208),VALUE(18)),
 KV(PASSAGE_SPAN(142,143), VALUE(14)))
**Result:** ARGMAX(KV('Matt Leinart', 78),KV('Cassel', 76)) = 'Matt Leinart' |
| | **Predicted Program (-sorting):** PASSAGE_SPAN(82,84) **Result:** Matt Cassel |
| ... Carney got a 38-yard field goal ... with Carney connecting on a **39-yard field goal** ... | **Question:** How many yards was the longest field goal? |
| | **Predicted Program:** MAX(VALUE(14),VALUE(11))
**Result:** MAX(39, 38) = 39 |
| | **Predicted Program (-sorting):** VALUE(11) **Result:** 38 |

(b)

Table 6: Examples of counting and sorting questions on DROP development set, where NeRd with the corresponding operations gives the correct predictions, while the variants without them do not. **(a)**: counting; **(b)**: sorting.

| | EM | F1 |
|---|---|---|
| Hard EM
with thresholding | **80.58** | **83.42** |
| Hard EM | 73.72 | 77.46 |
| Maximum Likelihood | 63.96 | 67.98 |

Table 7: Results of different training algorithms on DROP development set. For each setting, we present the best results on the development set.

| | Accuracy |
|---|---|
| Seq2prog | 51.9 |
| Seq2prog+cat | 54.2 |
| NeRd | **79.7** |
| NeRd (-pretraining) | 71.6 |
| NeRd (20%) | 56.5 |

Table 8: Results on MathQA test set, with NeRd and two variants: (1) no pre-training; (2) using 20% of the program annotations in training.

the models, such as SQuAD (Rajpurkar et al., 2016), CoQA (Reddy et al., 2019), GLUE (Wang et al., 2019), etc. A bunch of representative models are proposed for these benchmarks, including BiDAF (Seo et al., 2017), r-net (Wang et al., 2017), DrQA (Chen et al., 2017), DCN (Xiong et al., 2016) and QANet (Yu et al., 2018). More recently, massive text pre-training techniques, e.g., ELMo (Peters et al., 2018), BERT (Devlin et al., 2019), XLNet (Yang et al., 2019) and Roberta (Liu et al., 2019), have achieved superior performance on these tasks. However, for more complicated tasks that require logical reasoning, pre-trained models alone are insufficient.

On the other hand, semantic parsing has recently seen a lot of progress from the neural symbolic approaches. Jia & Liang (2016); Dong & Lapata (2016); Zhong et al. (2017) applied neural sequence-to-sequence and sequence-to-tree models to semantic parsing with full supervision. Liang et al. (2017); Neelakantan et al. (2016); Krishnamurthy et al. (2017); Guu et al. (2017); Liang et al. (2018) have advanced the state-of-the-art in weakly supervised semantic parsing on knowledge graphs and tabular databases. However, most of the successes of semantic parsing are limited to structured data sources. In contrast, our work naturally extends the complex reasoning in semantic parsing to reading comprehension by introducing the span selection operators. Several methods for training with weak supervision have been proposed in the context of weakly supervised semantic parsing including Maximum Marginal Likelihood (Berant et al., 2013; Krishnamurthy et al., 2017; Dasigi et al., 2019; Guu et al., 2017), RL (Liang et al., 2017; 2018) and Hard EM (Liang et al., 2017; Min

et al., 2019). Our approach is based on Hard EM due to its simplicity and efficiency, and extends it by adding a decaying threshold, which improves its robustness against spurious programs.

In the broader context, neural symbolic approaches have been applied to Visual Question Answering (Andreas et al., 2016; Mao et al., 2019; Johnson et al., 2017), where the neural architecture is composed with sub-modules based on the structured parses of the questions. Another line of work studied neural symbolic approaches to learn the execution of symbolic operations such as addition and sorting (Graves et al., 2014; Reed & de Freitas, 2016; Cai et al., 2017; Dong et al., 2019). In this work, we study neural symbolic approaches for reading comprehension tasks that require discrete reasoning over the text (Dua et al., 2019; Hu et al., 2019; Andor et al., 2019; Amini et al., 2019).

## 6 CONCLUSION

We presented the Neural Symbolic Reader (NeRd) as a scalable integration of distributed representations and symbolic operations for reading comprehension. NeRd architecture consists of a reader that encodes text into vector representation, and a programmer that generates programs, which will be executed to produce the answer. By introducing the span selection operators, our *domain-agnostic* architecture can generate *compositional* programs to perform complex reasoning over text for different domains by only extending the set of operators. We also overcome the challenge of weak supervision by applying data augmentation techniques and hard EM with thresholding. In our evaluation, using the same model architecture without any change, NeRd significantly surpasses previous state-of-the-arts on two challenging reading comprehension tasks, DROP and MathQA. We hope to motivate future works to introduce complex reasoning to other domains or other tasks in NLP, e.g., machine translation and language modeling, by extending the set of operators.

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

## A    MORE DETAILS ABOUT THE INPUT PREPROCESSING

We preprocess the input passages and questions in a similar way as the input preprocessing of DROP dataset described in (Andor et al., 2019). Specifically, to facilitate the usage of BERT, we split up the documents longer than $L = 512$ tokens. Meanwhile, we extract the locations and values of the numbers, so that they can be retrieved via indices when applying numerical operators. We apply the same input preprocessing on MathQA as well.

## B    MORE DISCUSSION ABOUT THE DOMAIN SPECIFIC LANGUAGE

To better support numerical reasoning, sometimes we need to leverage pre-defined constants for our computation. On MathQA, we have shown that applying the constant 3600, which is provided in their pre-defined question-agnostic constant list, is necessary for the calculation in Table 3. Meanwhile, we find that defining such a constant list is also helpful on DROP benchmark. For example, a variant of the sample numerical operation question in Table 2 is "How many people, in terms of percentage, were not either solely white or solely African American?", and such questions are included in DROP dataset as well. In this case, unless we are able to use the number 100 in our calculation, there is no way to obtain the correct answer. Again, previous works design specialized modules to deal with such questions, which is the main role of the negation module illustrated in Figure 1. On the contrary, we introduce a constant list that is callable for every question, so that the model can learn to apply any constant covered in the list, without the need of manually designing separate modules for questions requiring different constants.

In our evaluation, for DROP, we used $[100, 12, 28, 29, 30, 31, 1, 0]$ as the constant list, which is helpful for percentage and date time calculation. For MathQA, we used the constant list provided in their public dataset, which includes 23 constants that cover common conversion between different units, domain-specific constants for geometry, physics and probability, etc.

## C    MORE DETAILS ABOUT THE MODEL ARCHITECTURE

### C.1    READER

The reader implementation is largely the same as (Andor et al., 2019). Specifically, for the embedding representation of the reader component, we feed the question and passage jointly into BERT, which provides the output vector of each input token $t_i$ as $e_i$. Unless otherwise specified, the encoder is initialized with the uncased whole-word-masking version of BERT$_{\text{LARGE}}$. We denote the size of $e_i$ as $H_0$.

### C.2    PROGRAMMER

The core architecture of the programmer is a 1-layer LSTM with the hidden size of $H = 512$. To formally describe the input space and output space of the programmer, we denote $R$ as the size of the reserved tokens, which include both operators and constants in a domain-specific language, and the special start and end tokens $[\text{GO}]$ and $[\text{EOF}]$; and $L = 512$ as the total number of the question and passage tokens in a single sample. Samples with fewer than $L = 512$ tokens will be padded with $[\text{EOF}]$ tokens to achieve this length. In the following, we discuss the details of each component.

**Input embedding.**    At each timestep, the programmer could generate a program token from: (1) the reserved tokens of the domain-specific language; and (2) the input question and passage tokens. The embedding of the $i$-th reserved token is

$$hr_i = E_r^T r_i$$

Where $E_r$ is a trainable embedding matrix of size $R \times H$, and $r_i$ is the one-hot encoding of the token.

For the $i$-th token in the input question and passage token list, their embedding is

$$ht_i = P_t e_i$$

Where $P_t$ is a trainable projection matrix of size $H \times H_0$.

**Attention module over the input.** At each timetstep $T$, let $[p_1, p_2, ..., p_{T-1}]$ denote the list of program tokens that are already generated in previous timesteps, and we define $[hp_0, hp_1, hp_2, ..., hp_{T-1}]$ as the *decoder history*, where $hp_0$ is the embedding vector of the `[GO]` token calculated as above; $[hp_1, hp_2, ..., hp_{T-1}]$ are $H$-dimensional vectors corresponding to the generated program token list, and we will discuss how they are computed later.

Denote $(h_T, c_T) = \text{LSTM}(hp_{T-1}, (h_{T-1}, c_{T-1}))$ as the hidden state of the LSTM decoder at timestep T, where $(h_0, c_0)$ is the trainable initial state, and $hp_{T-1}$ is the LSTM input.

For each of $hp_i$ in the decoder history, we compute

$$vh_i = W_h hp_i$$

Where $W_h$ is a trainable matrix of size $H \times H$.

The attention weight of each $hp_i$ in the decoder history is computed as

$$wh_i = \frac{\exp(h_T^T vh_i)}{\sum_{j=0}^{T-1} \exp(h_T^T vh_j)}$$

The attention vector of the decoder history is thus

$$att_h = \sum_{i=0}^{T-1} wh_i \cdot hp_i$$

This formulation is similar to the attention mechanism introduced in prior work (Bahdanau et al., 2014). Correspondingly, we compute the attention vector of the passage tokens $att_p$, and the attention vector of the question tokens $att_q$.

Afterwards, we compute

$$v_T = W_v[att_h; att_q; att_p; h_T]$$

Where $W_v$ is a trainable matrix of size $H \times 4H$, and $[a; b]$ denotes the concatenation of $a$ and $b$.

**Program token prediction.** We compute another attention vector of the question tokens $att_q'$ in a similar way as above, but with a different set of trainable parameters. Then for each input token, we have

$$ht_i' = P'[ht_i; ht_i \circ att_q']$$

$$hr_i' = P'[hr_i; hr_i \circ att_q']$$

Where $P'$ is a trainable matrix of size $H \times 2H$, and $\circ$ is the Hadamard product.

Let $H_T'$ be a $(R + L) \times H$-dimensional matrix, where the first $R$ rows are $hr_i'$ for $0 \leq i < R$, and the next $L$ rows are $ht_i'$ for $0 \leq i < L$. Then we compute

$$w_T' = H_T' \cdot v_T$$

Where $w_{Ti}'$ denotes the weight of selecting the $i$-th token as the next program token. This design is similar to the pointer network (Vinyals et al., 2015).

Note that a valid program should satisfy the grammar constraints, for instance, those listed in Table 1 on DROP dataset. Therefore, we compute a mask $m_T$ as an $(R + L)$-dimensional vector, where $m_{Ti} = 1$ when the $i$-th token is a valid next program token, and $m_{Ti} = 0$ if it is invalid. In the following, we take the DROP dataset as the example, and list some sample rules for mask generation:

(1) At the beginning of the program generation, $m_{Ti} = 1$ iff the $i$-th token denotes an operator;

(2) When the previous generated program token $p_{T-1}$ is PASSAGE_SPAN, then $m_{Ti} = 1$ iff the $i$-th token is from the passage. Similarly, if $p_{T-1}$ is QUESTION_SPAN, then $m_{Ti} = 1$ iff the $i$-th token is from the question.

(3) As discussed in Appendix A, we preprocess the data to extract the locations and values of numbers in the input question and passage, thus we can leverage it to generate masks for numerical calculation operators. Specifically, when $p_{T-1} \in \{\text{DIFF}, \text{SUM}, \text{VALUE}\}$, $m_{Ti} = 1$ iff the $i$-th token is from the constant list, or a number from either the input question or the passage.

With the generated program mask, we compute

$$w_T = w'_T - C(1 - m_T)$$

Where $C$ is a large positive constant to ensure that the weight of an invalid program token is much smaller than the valid program tokens. In practice, we use $C = 1e6$. Such a grammar-based decoding process is a common practice in order to ensure the syntactic correctness of the generated programs (Krishnamurthy et al., 2017; Liang et al., 2017; Bunel et al., 2018).

Afterwards, the model predicts $p_T = \arg\max_i(w_T)$ as the next program token. We can also apply the beam search for decoding, but we find that the greedy decoding is already sufficient to provide good results, while the inference process is also much faster than the beam search.

Finally, $hp_T = H'_{T \, p_T}$ is the vector representation corresponding to $p_T$, which is appended to the decoder history for generating the next program token.

## D  More details about training

### D.1  Data augmentation

In this section, we discuss the details of our data augmentation process for counting and sorting questions on DROP. To obtain training samples for counting questions with ground truth annotations, starting from the span selection questions in the training set, we filter out those questions that either can be answered by using the QUESTION_SPAN operation, or do not start with any interrogative in ["What", "Which", "Who", "Where"]. Afterwards, we replace the interrogative with "How many", and modify the ground truth program correspondingly. In this way, we can augment 15K additional questions for counting in DROP training set.

To annotate the key-value pairs, for each entity recognized by the CoreNLP tool, we search for the numbers that are in the same clause as the entity, i.e., not separated by any punctuation mark, and discard those entities that do not have any nearby number satisfying this constraint. Afterwards, we filter out those questions that do not include any superlative in ["longest", "shortest", "largest", "smallest", "most" and "least"]. For the remaining questions, we call each of the sorting operations, i.e., ARGMAX, ARGMIN, MAX, MIN, with all extracted key-value pairs as the arguments. For ARGMAX and MAX operators, the key-value pairs are sorted in the descending order of their values; for ARGMIN and MIN operators, they are sorted in the increasing order of their values. If any of the resulted sorting program yields the correct answer, the program is included into the training set. In this way, we can annotate 0.9K questions using ARGMAX or ARGMIN operations, and 1.8K questions using MAX or MIN operations in DROP training set.

### D.2  Training configuration

For the training algorithm described in Algorithm 1, the initial threshold $\alpha_0 = 0.5$, and the decay factor $\gamma = 0.5$. We perform early stopping when both exact match and F1 score on the development

| Passage | Question | Ground truth |
|---|---|---|
| ... but had to settle for a **23-yard field goal by kicker Matt Bryant** ... | How many field goals shorter than 30 yards did Matt Bryant kick? | 3 |
| ... from a sample of 40 Sherman tanks, **33 tanks burned** (82 percent) and **7 tanks remained unburned** ... | How many more Sherman tanks burned out than survived in the Normandy Campaign? | 22 |

Table 9: Some samples in DROP training set with the wrong annotations, which are discarded by NeRd because none of the annotated programs passes the threshold of our training algorithm.

| Question type | Passage | Question | Prediction |
|---|---|---|---|
| Question span | The **campaigns of 1702 and 1703** showed his limitations as a field officer... In early 1704 , he spoke with the envoy of Savoy about possible opportunities in their army ... | What happened first, the Hague campaigns as field officer or he spoke with envoy of Savoy for opportunities in the army? | **Prediction:** QUESTION_SPAN(7,10) **Result:** "campaigns as field officer" **Ground truth:** "campaigns of 1702 and 1703" |
| Counting | ... The five regions with the lowest fertility rates were Beijing (0.71), **Shanghai (0.74)**, **Liaoning (0.74)**, Heilongjiang (0.75) ... | How many areas had a fertility rate of .74? | **Prediction:** COUNT( PASSAGE_SPAN(216, 216), PASSAGE_SPAN(223, 223), PASSAGE_SPAN(230, 231)) **Result:** COUNT(''Beijing", "Shanghai", "Liaoning") = 3 **Ground truth:** 2 |
| Sorting | ... to set up Nugent's career-long **54-yard field goal** to give the Jets a 9-3 lead ... The half ended when Brown came up five yards short on a 59-yard field goal attempt ... | How many yards was the longest field goal? | **Program:** MAX(VALUE(16), VALUE(20)) **Result:** MAX(54, 59) = 59 **Ground truth:** 54 |

Table 10: Examples of wrong predictions on DROP dev set.

set do not improve for two consecutive training iterations. For both DROP and MathQA datasets, the training typically takes around $50K \sim 60K$ training steps.

For both tasks in our evaluation, we train the model with Adam optimizer, with an initial learning rate of 5e-5, and batch size of 32. Gradients with $L_2$ norm larger than 1.0 are clipped.

# E    EXAMPLES OF WRONG ANNOTATIONS ON DROP

Table 9 lists some examples of wrong annotations in DROP training set. Specifically, the first annotation is wrong because the crowd worker simply counts the number of field goals included in the entire passage, without considering the constraints of lengths and the kicker's name; on the other hand, the second mistake comes from the wrong numerical calculations. For both samples, the highest likelihood among all programs with the annotated answer is smaller than 1e-4, thus are not included during training, which is why the thresholding helps significantly.

# F    EXAMPLES OF WRONG PREDICTIONS ON DROP

Table 10 presents some error cases of NeRd on DROP development set.

