# OpenReview forum: "Neural Symbolic Reader: Scalable Integration of Distributed and Symbolic Representations for Reading Comprehension"
_ICLR.cc/2020/Conference — Accept (Spotlight)_

### Official Review · AnonReviewer2 · 2019-10-09
**Official Blind Review #2**

**Rating:** 8

**Review:**

This paper presents a semantic parser that operates over passages of text instead of a structured data source.  This is the first time anyone has demonstrated such a semantic parser (Siva Reddy and several others have essentially used unstructured text as an information source for a semantic parser, similar to OpenIE methods, but this is qualitatively different).  The key insight is to let the semantic parser point to locations in the text that can be used in further symbolic operations.  This is excellent work, and it should definitely be accepted.  I have a ton of questions about this method, but they are good questions.  The rest of this review focuses on things that I thought could be more clear, or that raise new questions, and might sound negative.  Please understand them, however, in terms of my overall score and what I said above.

The three claimed contributions are (1) better numbers, (2) better compositionality / domain applicability, and (3) better interpretability.

(2) and (3) sound a bit like overclaiming in the introduction to me, as there isn't a whole lot of nested composition in the language used by NeRd, and the BERT calculator in principle is almost as compositional and interpretable (also, e.g., NAQANet can add and subtract an arbitrary number of numbers, also, and it tells you which ones they are, just as NeRd does).  Later in the paper the specifics of those claims are made more clear, and while they are justified, they are very narrow claims.  To me, someone who is intimately familiar with this research area, the key contributions (the things that I learned) are (1) using passage-span and key-value predicates actually works, (2) how much difference hard EM and thresholding make, and (3) the data augmentation in this work is pretty clever.  (2) was intuitively clear to me after seeing Dasigi's iterative search paper and Min's hard EM paper, but the difference in results presented here is pretty striking.

Compositionality:

The authors claim that their method is compositional and domain agnostic, while all previous methods had hand-crafted modules for specific question types.  However, I see little reason to believe there's much of a difference here.  You also defined operations that are tailored to the dataset, and are basically identical to the operations that others have used.  I see no evidence that NeRd actually generalizes to program types that are beyond what is captured by other methods.  It's possible that this happens, but there is no evaluation that discusses this, and from all of the examples I'm led to believe that this is basically also just learning a few program templates, the same ones learned by previous methods.  With the weak supervision that you have, are you actually able to find more complex programs during your search?  Some kind of demonstration of actual compositionality on the more complex questions in DROP would make a very strong argument for your claims; without that, they ring a little hollow.

Interpretability:

The use of passage-span as a predicate is really interesting, and it raises a lot of questions.  This predicate lets the model shortcut any interpretable reasoning and do operations entirely inside the encoder/parser.  For example, your first example in table 2 ostensibly requires filtering the numbers in the passage to those that are percentages associated with groups, then filtering them again to those where the percentage is larger than 16, then returning the associated groups. But your method jumps straight to returning a set of passage spans.  This is hardly interpretable.  (In fairness, no prior method provides interpretable reasoning for this kind of operation either.)  But the fact that you have this predicate lets the model do these filters and greater-than comparisons inside the network in an opaque way, while also getting interpretable operations for some questions (table 5 is further confirmation of this, and of the fact that you probably are not capturing many of more the complex, compositional questions in DROP).  But how does the network decide which to do?  Any argmax or max question, and many count questions, could be answered by passage-span alone.  With only weak supervision, and with the parser having the ability to shortcut these more interpretable operations, how often are you actually getting the interpretable one, and what's causing the model to choose it?

Similarly, how often does an argmax or a max operation actually operate on the full set that you would expect it to?  Or does it just do the argmax internal to the network, and output only one item as an argument to the argmax?  If the later, this again hurts your claims of better interpretability over prior methods, as the logic is just as opaque as before.  This also seems like a really hard search problem in how you've set up your DSL - what would make your search over programs actually select all of the correct arguments?  Because you're selecting passage spans directly instead of performing some kind of matching operation, you have to have your search select all of the appropriate spans for this to be "interpretable", and not just hiding the logic inside of the network.  But that seems like a totally intractable search.  You found a clever way to get around this for count questions (even though that still implicitly hides a bunch of filtering logic, as noted above), but I don't know how to make it work for maxes and argmaxes.

Another question raised by the passage-span predicate: the more you use bare passage-span programs for training, the more the network learns to put all of its compositional reasoning inside, in an opaque way, instead of giving you interpretable compositionality.  At one extreme, you end up with something like NABERT (or even less compositional), where basically everything is inside the network.  At the other extreme, where you don't have passage-span, you are left with a crippled semantic parser that can't handle most of the questions.  But using the predicate introduces tension in the model between interpretability and flexibility.  How do we resolve this tension?  (This isn't something I expect your paper to address, it's just a really interesting and important question raised by your work.)

Parser:

Prior work has found benefit in using runtime constraints on parser outputs, or grammar-based decoding.  It looks like you are doing neither of those, yet you're able to output specific token indices and number indices in your programs.  Are you really not doing anything special to handle those?  How does the decoder know token indices?  I feel like something must be missing here, or a simple LSTM decoder is more magical than I thought.

Evaluation:

Why only show results on DROP dev, and not on the test set?  It's possible that your higher numbers are because you were better able to overfit to the dev set, which you presumably used during training.  (I don't think that that's likely, but it's a concern that would be easily avoided by evaluating on test.)

**Experience Assessment:**

I have published in this field for several years.

**Review Assessment: Checking Correctness Of Derivations And Theory:**

N/A

**Review Assessment: Checking Correctness Of Experiments:**

I carefully checked the experiments.

**Review Assessment: Thoroughness In Paper Reading:**

I read the paper thoroughly.

---

> ### Author Response · Authors · 2019-11-15
> **Clarification and more discussion**
>
> Thanks for appreciating our work and your insightful comments! We respond to your questions below in terms of Contributions, Compositionality, Interpretability, Parser details and Evaluation.
>
> Clarifications on contributions
>
> (1) We would like to clarify that we did not claim the the model is fully interpretable, solves all the compositional questions in DROP, or requires no domain-specific languages. Instead, we are arguing that NeRd is *better* than previous methods in these dimensions, so we scoped all the claims in comparison with previous methods in the paper.
>
> (2) We would like to emphasize that the claims, especially the ones on compositionality and easier domain adaptation, are supported by considering *both the experiments on DROP and MathQA*. We have demonstrated that the semantic parsing approach can be applied over text to achieve good performance on DROP, but due to the challenges of searching for highly compositional programs on DROP (more on this in "Compositionality" discussion), the ability of NeRd to generate highly compositional programs are better demonstrated on MathQA. And easier domain adaptation is shown by the fact that we applied the same architecture to MathQA using the DSL released by the author of MathQA without any further efforts in DSL or architecture design.
>
> (3) We would like to point out some important differences with previous methods:
>
> 1. NeRd, which operates over text, is similar to other semantic parsers in that it has the expressive power to generate compositional programs by recursively calling the operators, and uses a single program decoder, e.g., LSTM in our evaluation, to answer different types of questions. In contrast, previous methods on DROP rely on designing specialized neural modules to answer different types of questions, and more importantly, those modules cannot be applied recursively or compositionally.
>
> Let’s take arithmetic as an example, and show how previous approaches would require specialized neural modules to handle it. BERT with calculator has to introduce an operator "Sum3" that selects three arguments to support summing up 3 numbers, and extending that approach to handle numerical operations with an arbitrary number of arguments would require creating an exponential number of operators like "Sum4", "Sum2Diff1", etc; on the other hand, these operations can be naturally handled by recursively calling "Sum" and/or "Diff" a few times with a model that supports generating compositional programs such as NeRd. MTMSN designs a specialized “negation” module, which predicts whether a number should be subtracted from 100 to handle negation questions in percentage calculation, e.g., “How many percent were not German”, because this type of questions appear frequently in the DROP dataset. As you pointed out, NAQANet can add or subtract arbitrary number of numbers by performing a 3-class classification for each number in the passage, representing plus, minus or zero; this design choice is also applied in several subsequent approaches on DROP, e.g., NABERT and MTMSN. However, if the questions require more math operators,like in MathQA, this approach cannot easily support complex computations interleaving different types of operators, e.g., ((1+2)*(3-5)+5^2/6), while such questions can be naturally supported by the compositional programs generated by NeRd in our MathQA experiments.
>
> In a word, we agree that if we solely focus on the performance, all these different approaches work reasonably well for arithmetic questions in DROP since they are specially designed for it; however, none of them can be directly adapted to MathQA, which requires highly compositional arithmetic reasoning and a much larger set of operators. By providing the model with the capability of generating compositional programs, it is easier to adapt NeRd to other domains than other methods compared in our work. These compositional programs are already commonly used in semantic parsing to naturally support complex reasoning without designing specialized neural modules for different types of questions. Our contribution, as you noted, is to show that we can apply them to unstructured input as well, i.e., text, to achieve the same adaptability and compositionality.

---

> > ### Author Response · Authors · 2019-11-15
> > **Clarification and more discussion (part 2 of Contribution discussion, discussion on Compositionality and domain agnostic)**
> >
> > 2. For counting questions, by calling span selection operators within counting, the programs generated by NeRd aims to select spans related to the question, and count the number of them to provide the answer; on the other hand, previous methods typically treat counting as a multi-class classification problem over 0-9. We agree that for counting and sorting, the selected spans may not be what we desire, and it could still hide part of reasoning inside the network, e.g., why the model considers the selected spans to be related to the question, but it is already more interpretable than providing the final answer alone. Take counting as an example, we would argue that providing the answer “3” accompanied with three spans selected is more interpretable than predicting a number “3” alone. Please refer to Table 6 in the paper for more examples.
> >
> > We believe these differences show that NeRd, although still far from fully interpretable, is one step forward and demonstrates better interpretability than previous approaches.
> >
> >
> > Compositionality and domain agnostic:
> >
> > NeRd can generate nested expressions or compositions like any semantic parsers (also demonstrated in MathQA experiments). We agree that the current programs generated on DROP are not as compositional as other semantic parsing tasks. However, this is not due to the limitation of the model's expressive power, but due to the difficulty in searching, which you also noted in the comments, for highly compositional gold programs.
> >
> > For example, for compositional questions like "How many more touchdowns did Jonathan Stewart get compared to Adrian Peterson?", which requires applying COUNT twice and then a DIFF, searching for the gold program requires resolving two challenges posed by weak supervision: (1) it has to explore an exponentially large search space, since the program includes many operations, and in particular, for counting operations, the space of possible spans to select is very large, as discussed in Section 3.1 in the paper; (2) there could be a huge amount of spurious programs that return the same answer, which makes it hard for the model to learn from the correct program. These two challenges are the main obstacles for the model to learn to generate highly compositional programs given only weak supervision from the ground truth answer. It is definitely an interesting future direction to address this search problem, for example, building upon existing curriculum learning and iterative search method [Dasigi et al, 2019].
> >
> > On the other hand, when a small number of compositional programs are available during training, NeRd can learn to generate such compositional programs for unseen questions. For example, in MathQA, even when only 20% of the programs are given, NeRd is able to generate highly compositional programs, with more than 10 levels of compositions, for unseen questions; meanwhile, other methods, as discussed in "clarifications" section above, often require specialized designs for DROP, and thus cannot easily adapted to MathQA or other domains.
> >
> > As described in the abstract, or claim of “domain-agnostic” means “the same neural architecture works for different domains", i.e., the *architecture* doesn't need dataset-specific crafting, and can remain the same for different domains. This is supported by the fact that we used the same architecture for both DROP and MathQA and obtained strong results on both benchmarks. We agree that it still relies on different operators to work on different domains, but we would argue that avoiding domain-specific languages is a grand challenge, unless we can design a very general language that captures all the common sense reasoning, which is arguably the most important and challenging problem in symbolic AI. Also, in some cases the domain specific language is given as part of the task specification, e.g., in MathQA, WikiSQL, etc.

---

> > > ### Author Response · Authors · 2019-11-15
> > > **Clarification and more discussion (Interpretability)**
> > >
> > > Interpretability:
> > >
> > > As you noted, there is a trade-off between interpretability and flexibility/accuracy, and we kept the span selection for the sake of flexibility/accuracy. We agree that by enabling the model to simply call the basic span selection operator to answer related questions, this may hurt the interpretability and push part of the reasoning into the network, but we are not able to avoid it in order to get a good accuracy. Answering the questions in a fully interpretable way would require designing more advanced operators as well as solving the search problem (more discussion can be found in the "Compositionality" section above), and we leave it as important future work.
> > >
> > > Although NeRd showed better interpretability on counting compared to previous methods, despite that it might hide some reasoning inside span selection, we agree that sorting is much more challenging due to the difficulty in search and data augmentation. For the specific case on argmax/max/argmin/min, there is no guarantee that the program will select all key-value pairs desired as arguments for the following reasons: (1) as stated in Appendix D.1, our data augmentation itself is able to extract programs with these operators only for a small proportion of questions in the training set and some important entities could be missed, because while the CoreNLP is a powerful tool to extract entities in the passage, it is not always successful and accurate; (2) the augmented programs themselves include noise in the sense that they may also miss relevant key-value pairs, and/or include unrelated ones as the arguments. The intuition is that only the key-value pair containing the ground truth answer should clearly be included, but for any other key-value pair, there is no perfect automatic way to determine whether it should be included or excluded, thus can result in noise. For example, for a question, "how long is the longest touchdown in the first quarter?", we only know that the key-value pair with the same value as the annotated number in the ground truth answer should be included (assuming the ground truth answer only appears once for now), and for any other key-value pair, there is chance that the number is not from the first quarter, which can be viewed as a kind of spurious programs, thus introduces noise. In our experiments, what we saw is that including more than one arguments for programs using these operators in training data augmentation helps the model better answer such questions, and the model indeed learns to select more than one key-value pairs with the largest/smallest values as the arguments. We find that these operators help when the largest/smallest numbers are close in the passage, as the samples shown in Table 6. Our intuition is that the network itself is able to differentiable between values that are far apart within the network, e.g., 50 and 1; however, when the top numbers are close, it is harder for the network to differentiate [Wallace et al, 2019], in which case the symbolic operator can help.
> > >
> > > In general, finding the right trade-off between interpretability and accuracy/flexibility is challenging. As your comment pointed out, one of the fundamental questions in any neural symbolic approaches for reasoning is how to divide the labor between the "implicit" black-box reasoning within the neural network and the "explicit" reasoning in the symbolic representation. In this work, we kept the span selection operators for the sake of flexibility/accuracy, and tried to push more reasoning into the programs for counting and sorting as a step forward in improving both interpretability and accuracy, though we agree that it is still far from ideal. We defer adding more advanced operators, such as filtering suggested by the reviewer, as promising future work.
> > >
> > > Eric Wallace, Yizhong Wang, Sujian Li, Sameer Singh, and Matt Gardner. "Do NLP Models Know Numbers? Probing Numeracy in Embeddings." EMNLP 2019.

---

> > > > ### Author Response · Authors · 2019-11-15
> > > > **Clarification and more discussion (Parser and Evaluation)**
> > > >
> > > > Parser:
> > > >
> > > > We extended Appendix C to provide the full model details about how the programs are generated. In fact, we did perform the grammar-based decoding to ensure that the generated programs are syntactically valid. Specifically, the program generation process is similar to the pointer network [Vinyals et al, 2015], where we compute the attention weights over all valid possible program tokens at each timestep, and select the one with the highest prediction probability. Note that we preprocess the input passage and questions to extract the location and value of all numbers included, as discussed in Appendix A, which is used to generate the program mask so that the arithmetic operators only take numbers as the argument.
> > > >
> > > > Oriol Vinyals, Meire Fortunato, and Navdeep Jaitly. "Pointer networks." Advances in Neural Information Processing Systems. 2015.
> > > >
> > > > Evaluation:
> > > >
> > > > Due to time constraint, we only evaluated locally on dev set at the time of submission. We submitted to the DROP server later, and the test result is better (+1.18% on F1 and +1.37% on Exact Match) than other baselines.

---

### Official Review · AnonReviewer1 · 2019-10-24
**Official Blind Review #1**

**Rating:** 6

**Review:**

This paper discusses an extended DSL language for answering complex questions from text and adding data augmentation as well as weak supervision for training an encoder/decoder model where the encoder is a language model and decoder a program synthesis machine generating instructions using the DSL. They show interesting results on two datasets requiring symbolic reasoning for answering the questions.

Overall, I like the paper and I think it contains simple extensions to previous methods referenced in the paper enabling them to work well on these datasets.

A few comments:

1 - Would be interesting to see the performance of the weak supervision on these datasets. In other words, if the heuristics are designed to provide noisy instruction sets for training, we need to see the performance of those on these datasets to determine if the models are generalising beyond those heuristics or they perform at the same level which then may mean we don't need the model.

2 - From Tab 4, it seems the largest improvements are due to span-selection cases as a result of adding span operators to the DSL. A deep dive on this would be a great insight (in addition to performance improvement statements on page 7).

3 - Since the span and value operators require indices to the text location, could you please clarify in the text how that is done? Do LSTMs output the indices or are you selecting from a preselection spans as part of preprocessing?



**Experience Assessment:**

I have read many papers in this area.

**Review Assessment: Checking Correctness Of Derivations And Theory:**

I carefully checked the derivations and theory.

**Review Assessment: Checking Correctness Of Experiments:**

I carefully checked the experiments.

**Review Assessment: Thoroughness In Paper Reading:**

I read the paper thoroughly.

---

> ### Author Response · Authors · 2019-11-15
> **Clarification and discussion**
>
> Thank you for your valuable comments, and we are glad that you like our paper! About your comments:
>
> 1. The heuristics used to obtain training data augmentations from weak supervision all rely on access to the ground truth answer, so they cannot be directly applied for prediction given just questions and passages. In other words, the heuristics are only applicable during training, so at the inference time, the model has to generalize to answer unseen questions.
>
> 2. The span selection operators are key components in the DSL because (1) they can be used to answer the single span and multiple span questions; (2) they can support higher level reasoning such as counting and sorting.
>
> For the significant improvement on multiple-span questions, we hypothesize that it is because the same PASSAGE_SPAN operators are used in both single-span, multiple-span as well as counting and sorting questions; i.e., multiple-span questions are answered by calling PASSAGE_SPAN a few times, and counting also calls PASSAGE_SPAN to find the spans to count. Therefore, the model obtains more training signals to learn how to use such operators.
>
> 3. We extended Appendix C to provide the full details of how the programs are generated. Specifically, the program generation process is similar to the pointer network [Vinyals et al, 2015], where we compute the attention weights over all valid possible program tokens at each timestep, and select the one with the highest prediction probability.
>
> Oriol Vinyals, Meire Fortunato, and Navdeep Jaitly. "Pointer networks." Advances in Neural Information Processing Systems. 2015.

---

### Official Review · AnonReviewer3 · 2019-10-25
**Official Blind Review #3**

**Rating:** 6

**Review:**

-- define cascade errors when you first use the phrase
-- basic english grammar could be fixed but is not interfering with understanding
-- what is the early stopping criterion in Alg 1?
-- did you try any other values for the initial threshold and decay factor?
-- At the end of Section 4.1 DROP: ", but not the same." - I can't parse what this last clause is supposed to mean.
--the diff sum example in table 2 is confusing; the program appears to sum up the numbers but the result is a subtraction without a sum operation in it. Would be clearer to show the sum in the result line as well rather than distribute the subtraction. Also, shouldn't it be diff(9, sum(10, 12))?
-- I think you should pull at least some commentary about the constant used in Table 3 from Appendix B and include it in the main paper (or at least mention Appendix B is the place to look). Can you add a table in an appendix showing the complete list of operators used?
-- Nice results in Table 4 on the dev set. Are there Test set results as well?
-- The organization of the Baselines 4.3 section and the Results 4.4 is confusing. For example, you mention that you test different variants of NeRd, operator variants, and mathqa, but then the results are not mentioned for these experiments until the next page. I found myself immediately looking for the numbers/results when you introduce the experiment. I would pair your experiment description with the results rather than grouping all experiment descriptions and then grouping all results, especially when the order of the experiment descriptions does not match the order of the results presented. For example, in baselines you discuss training variants and then operators. Then in Results you discuss operators before variants. It is too disconnected and makes the reader jump around a bunch. Same goes for the drop baselines where you mention a bunch of models, and I would prefer the Results/discussion paired with each one, rather than having to wait for it down below.
-- Overall it seems like a solid work; good empirical results showing improvements of each purported contribution. The model itself is a relatively simple construction of basic component, but the combination with the DSL is intuitive and makes sense. I don't think the novelty in model here is the main selling point anyways; the training variants and the demonstration of how well a DSL approach can do combined with previously introduced methods.
-- I find the model description to be slightly unclear. In Fig 1 for example there is an arrow that connects passage to compositional programs. What does that arrow represent? I think you should elaborate on how the attention over the encoded text interacts with the attention over previously generated tokens. Equations would make this far more explicit as is I am left with a lot of questions on how to implement your model. Maybe you can add to your appendix? Or release your code? That is mentioned either.

**Experience Assessment:**

I have published one or two papers in this area.

**Review Assessment: Checking Correctness Of Derivations And Theory:**

N/A

**Review Assessment: Checking Correctness Of Experiments:**

I carefully checked the experiments.

**Review Assessment: Thoroughness In Paper Reading:**

I read the paper thoroughly.

---

> ### Author Response · Authors · 2019-11-15
> **Clarification and revision**
>
> Thanks for your constructive feedback! We have incorporated your comments in our revision, and we respond to your questions below:
>
> -- define cascade errors when you first use the phrase
>
> We use "cascade error" to refer to errors caused by the previous phase in a pipeline. In our case, we use this term to refer to the errors caused by data preprocessing for semantic parsing approaches, which parses the text into structured representations using tools such as SRL. We revised the paper to add a description in the introduction, where we first use this phrase.
>
> -- basic english grammar could be fixed but is not interfering with understanding
> We have done another round of proof-reading and updated the paper accordingly.
>
> -- what is the early stopping criterion in Alg 1?
> We perform early stopping when both exact match and F1 scores on the development set do not improve for two consecutive training iterations. We have updated Appendix D.2 to make this point clearer.
>
> -- did you try any other values for the initial threshold and decay factor?
> We have also tried other values for the initial threshold and decay factor, and we found that the specific values do not matter much for the final results as long as they are in a good range. Specifically, values in [0.2, 0.5] work well for both the initial threshold and decay factor (we tried 1/5, 1/4, 1/3, 1/2 within this range). Values larger than 0.5 will slow down training, and values smaller than 0.2 will decrease the quality of the final model.
>
> -- At the end of Section 4.1 DROP: ", but not the same." - I can't parse what this last clause is supposed to mean.
>
> We have rephrased the sentence to be "F1 score, which gives partial credits to a prediction that is not exactly the same as the ground truth, but overlaps with it”. For example, if the ground truth is "Barrack Obama", and the prediction is "Obama", the exact match score will be zero, while the F1 score is larger than zero.
>
> --the diff sum example in table 2 is confusing.
>
> Thanks for catching the problem! It is indeed a typo. We have fixed it in the revision, and modified the explanation accordingly to make it clearer.
>
> -- Discussion of the complete operator and constant lists.
>
> For DROP, we have included the complete constant list in Appendix B. Note that MathQA covers a wide range of mathematical questions, and in their public dataset, they released the complete lists with 58 operators and 23 constants, which could be too long to include in the paper. Therefore, we added a description in Appendix B, and refer to their paper and public dataset for complete details.
>
> -- Nice results in Table 4 on the dev set. Are there Test set results as well?
>
> Due to time constraint, we only evaluated locally on dev set at the time of submission. We submitted to the DROP server later, and the test result is better (+1.18% on F1 and +1.37% on Exact Match) than other baselines.
>
> -- The organization of the Baselines 4.3 section and the Results 4.4 is confusing...
>
> Thanks for the suggestions! We swapped the descriptions of operator variants and training variants in Section 4.3, so that the order in Section 4.3 matches the order in Section 4.4 now. The main reason why we did not pair the descriptions with results is for paper space arrangement. Given that our paper has several tables of different sizes, separating them out is sometimes sub-optimal in terms of space usage. Therefore, we tentatively keep the section organization as it is in this revision; however, in our camera ready version, we will try to re-organize these two sections if it looks better.
>
> -- Overall it seems like a solid work; good empirical results showing improvements of each purported contribution...
>
> Thanks for appreciating our work! We agree that the neural architecture of each component in our model, i.e., BERT as the encoder and an LSTM with attention as the program decoder, is not our main contribution. Instead, as pointed out by Reviewer 2, the key novelty is "This paper presents a semantic parser that operates over passages of text instead of a structured data source. This is the first time anyone has demonstrated such a semantic parser". To achieve this, we made several technical contributions to address the challenges in designing such a model, for example, the introduction of span selection operators so that compositional reasoning can be applied over text.
>
> -- I find the model description to be slightly unclear...
>
> The arrow means that the program can directly select spans from the passage by calling the PASSAGE_SPAN operator with the indices of start and end token; on the contrary, existing neural semantic parsers require an additional structured parser. We have written the detailed equations in Appendix C, including the attention mechanism. We will open source our code with our camera ready version.

---

### Public Comment · ~Blind_Name2 · 2019-11-08
**Review and Doubts**

 Thanks for you work. I read the paper thoroughly and had two question.

1. I consider it as a combination of [1] and [2]. And the result does not have obvious advantages compared to [1].

2. I also find the model description to be slightly unclear.  And I doubt that your experiments are not reproducible from your unclear description. Could you release your code? I don't find your result on the leaderboard of drop as well.


[1]Giving BERT a Calculator: Finding Operations and Arguments with Reading Comprehension
[2]A Discrete Hard EM Approach for Weakly Supervised Question Answering

---

> ### Author Response · Authors · 2019-11-15
> **Clarification and revision**
>
> Thanks for your interest in our work!
>
> 1. We have compared extensively with previous methods such as BERT with calculator, MTMSN, etc, in our paper. To further clarify the differences, we copy part of our response to Reviewer 2 that emphasized the differences below.
>
> "We would like to point out some important differences with previous methods:
>
> NeRd, which operates over text, is similar to other semantic parsers in that it has the expressive power to generate compositional programs by recursively calling the operators, and uses a single program decoder, e.g., LSTM in our evaluation, to answer different types of questions. In contrast, previous methods on DROP rely on designing specialized neural modules to answer different types of questions, and more importantly, those modules cannot be applied recursively or compositionally.
>
> Let’s take arithmetic as an example, and show how previous approaches would require specialized neural modules to handle it. BERT with calculator has to introduce an operator "Sum3" that selects three arguments to support summing up 3 numbers, and extending that approach to handle numerical operations with an arbitrary number of arguments would require creating an exponential number of operators like "Sum4", "Sum2Diff1", etc; on the other hand, these operations can be naturally handled by recursively calling "Sum" and/or "Diff" a few times with a model that supports generating compositional programs such as NeRd. MTMSN designs a specialized “negation” module, which predicts whether a number should be subtracted from 100 to handle negation questions in percentage calculation, e.g., “How many percent were not German”, because this type of questions appear frequently in the DROP dataset. As you pointed out, NAQANet can add or subtract arbitrary number of numbers by performing a 3-class classification for each number in the passage, representing plus, minus or zero; this design choice is also applied in several subsequent approaches on DROP, e.g., NABERT and MTMSN. However, if the questions require more math operators,like in MathQA, this approach cannot easily support complex computations interleaving different types of operators, e.g., ((1+2)*(3-5)+5^2/6), while such questions can be naturally supported by the compositional programs generated by NeRd in our MathQA experiments.
>
> In a word, we agree that if we solely focus on the performance, all these different approaches work reasonably well for arithmetic questions in DROP since they are specially designed for it; however, none of them can be directly adapted to MathQA, which requires highly compositional arithmetic reasoning and a much larger set of operators. By providing the model with the capability of generating compositional programs, it is easier to adapt NeRd to other domains than other methods compared in our work. These compositional programs are already commonly used in semantic parsing to naturally support complex reasoning without designing specialized neural modules for different types of questions. Our contribution, as you noted, is to show that we can apply them to unstructured input as well, i.e., text, to achieve the same adaptability and compositionality."
>
> As for hard EM, in our ablation study (see Table 7 in the paper), we have shown that the thresholding technique we introduced significantly boosts the performance, thus is a new and crucial component in training.
>
> 2. We have extended Appendix C to include the full model details in our revision. We will release the code upon acceptance. Note that DROP leaderboard requires each published result to include the author information. To preserve the double-blind review process, we decided to publish the result after the paper decision.

---

### Author Response · Authors · 2019-11-15
**Response and revision**

We thank all reviewers for their insightful feedbacks and appreciating our work! We have revised the paper with the following major changes to incorporate the comments:

1. We update the appendix to include more discussion about the training configuration, domain-specific language in our evaluation, data augmentation, as well as the model architecture. In particular, to address reviewers’ confusion, we expand Appendix C to include the full details of the model.

2. We present the result on the hidden test set of DROP, and show that our NeRd model also outperforms other baselines.

---

### Decision · Program_Chairs · 2019-12-19

**Decision:**

Accept (Spotlight)

**Comment:**

Main content:

Blind review #1 summarizes it well:

This paper presents a semantic parser that operates over passages of text instead of a structured data source.  This is the first time anyone has demonstrated such a semantic parser (Siva Reddy and several others have essentially used unstructured text as an information source for a semantic parser, similar to OpenIE methods, but this is qualitatively different).  The key insight is to let the semantic parser point to locations in the text that can be used in further symbolic operations.  This is excellent work, and it should definitely be accepted.  I have a ton of questions about this method, but they are good questions.

--

Discussion:

The reviews all agree on a generally positive assessment, and focus on details that have been addressed, rather than major problems.

--

Recommendation and justification:

This paper should be accepted. Even though novelty in terms of fundamental machine learning components is minimal, but the architecture employing neural models to do symbolic work is a good contribution in a crucial direction (especially in the theme of ICLR).